# Green System Development in the Medinas of Tunis and Marrakesh—Green Heritage and Urban Livability

**Sarah Ben Salem \*, Chaima Lahmar, Marianna Simon and Kinga Szilágyi**

Institute of Landscape Architecture, Urban Planning and Garden Art, MATE University of Hungary, 1118 Budapest, Hungary; Lahmer.Chaima@hallgato.uni-szie.hu (C.L.); Simon.Marianna@szie.hu (M.S.); Mezosne.Szilagyi.Kinga@szie.hu (K.S.)
\* Correspondence: Ben.Salem.Sarah@phd.uni-szie.hu or sarah.ben.salem23@gmail.com

**Abstract:** Due to their authentic urban and architectural character, the Medinas of Tunis and Marrakesh became listed among the United Nations educational, scientific, and cultural organization (UNESCO) heritage sites in 1979 and 1985, respectively. Nowadays, the urbanization of the surrounding green areas and the climate change impacts on cities are degrading the Medinas' livability and their characteristic heritage. On the other hand, scientific knowledge and data about the green system in the dense urban cores of Medinas in the Maghreb region is still not a widely apprehended theme in the scientific domain. This research objective is to initiate nature-based and sustainable solutions in these cities by demonstrating the application of the urban green infrastructure (UGI) approach. As a research methodology, an analysis of the historical green system development in the Medinas is given to highlight their tangible and intangible values. The analysis goes over three periods: first, the medieval Islamic era, then the modern period during the French colony, and the contemporary city as a unique urban landscape. Finally, the study proposes a design guideline to prove the applicability of the UGI in the given historical morphologies by implementing the retained historical values of the historic green heritage in Medinas and the aspects of the site.

**Keywords:** urban green infrastructure; medieval Islamic cities; green system; green heritage; urban heritage; urban climate; Medinas of Tunis and Marrakesh; Islamic urban culture; innovation; livability



## 1. Introduction

Spatial-social dichotomy, increased use of motor vehicles, and changes in the networking and communication infrastructures, urban-rural, and core and periphery conflict were the predominant aspects of urban sprawl in the big urban agglomerations. As a consequence of these changes, most historic cities are located in the center of a growing metropolis and are affected by massive urban growth and tourism [1] (pp. 148–160). For the last two centuries, Maghreb cities have witnessed exponential demographic growth [2]. From the French colonization onwards, remarkable changes have occurred in the economic system and city-planning strategies [3].

Nowadays, the urban development has affected the historical medieval Islamic cities of Medinas from different aspects. In North Africa, the appellation of Medina is often given to Islamic historical cities—while the word Medina in Arabic language is the name of an urban settlement, and it is used in reference to the whole urban complex, or to its older part if it is physically differentiated from later additions [4] (p. 33).

In this study, the focus is given to the urban open spaces of Medinas, mainly the streets and alleys and other public places, which show degradation caused by urban heat island effects (rising temperature and degradation of mezzo climate). Some Medinas in the Maghreb region have been listed as world heritage protected sites. However, this protection can be adequate only if the territory can keep its inhabitants and the livability of the domain. This study reflects on nature-based and sustainable solutions by bringing forward the urban green infrastructure approach in the studied historical environments.

Since the late 1990s, the urban green infrastructure (UGI) concept has proved its ability to convey multiple benefits in different urban landscapes, mainly in Western Europe [5]. However, in other regions such as Africa, the concept of green infrastructure (GI) needs to be apprehended in theory and practice [6]. Due to the semi-arid geographical positions of the studied Medinas and the increasing urban heat effects over the last decades, the UGI concept should enhance urban health.

This research scope is to bring into front the problem of considering the improvement of the quality of public spaces in the Medinas in the present context and, on the other hand, to demonstrate the application of UGI in a historic site with a dense urban fabric. This paper introduces firstly a review on the green infrastructure approach applicability in urban cores and its multiple benefits. Then it presents the Islamic Medinas' urban development and the urban landscapes' consequences, thereafter, it discusses the urbanization issues in the Medinas, and finally suggests design guideline-based results.

## 2. Materials and Methods

The research methodology is based on case studies' analysis. The selected cities are examples of two historic sites with a particular dense organic urban structure. The study cases' analysis is not only based on literature and historic maps and graphs about urban and green system development through history and their connection to the city, but also, it highlights their tangible and intangible values. The analysis goes over three periods: first, the medieval Islamic era, then the modern period during the French colony, and the contemporary city as a unique urban landscape. Moreover, the research presents some recent social activities aiming to integrate green elements in the cities. As a result, this research proposes a design guideline to prove and demonstrate the possibility of applying the newly adapted strategy of UGI in the studied domains.

Since the study's objective is to demonstrate the applicability of UGI in a dense historical core, the paper firstly mentions a review on the UGI approach and states its important health benefits in the city. Thereafter, the paper follows a case study method. The selection of the case studies was based on their urban typologies and historic similarities. The study subjects are two historic Medinas. The first one is the Medina of Tunis. This Medina is the first listed heritage site of the United Nations educational, scientific, and cultural organization (UNESCO) among Medinas in North Africa [7]. The second case study from the Maghreb region is the Medina of Marrakesh. The selection of these two cities for a comparative study was based on the following reasons: Tunis is located in a typical maritime semi-arid climate, while Marrakesh is an arid area. The differences in period and position help us explore how Muslims adapted their cities' structure to different climatic conditions, and also, the standard adaptation features could be highlighted. Moreover, the Medina of Marrakesh was founded more than three centuries after the Tunis Medina. The integration of the green spaces in the two cities is then different. On the other hand, the two selected Medinas have similarities in their history; both cities were governed by "Almohades" and are UNESCO cultural heritage.

In the analysis of the urban development and transformation of the urban heritages section, the study aims to prove that the green spaces have decreased compared to the past. In addition to that, it emphasizes the particularities of each Medina's historical landscape, and the green elements that are part of the protected zones, to come out with conclusions related to the tangible and intangible values of the historic landscape elements, and to the strategies adapted by the city founders to integrate their cities in the immediate environment. These values and attributes, and the understanding of the original urban morphologies will be an asset for the study to come out with design points that can be adapted in the actual situation of the historic cities. The analysis is based on historical literature sources and historical maps about the urban and green system development of Medinas.

Moreover, an analysis is given to reflect on the current situation of the sites and to compare their urban cores and green areas ratios. Additionally, the study mentions the

recent urban strategies related to the integration of green zones in the historic sites, and discusses their feasibility. Besides, several examples of social actions aiming to improve the Medinas' livability were collected and evaluated. These scientific resources will help us to assess the actual situation in the Medinas. The research materials are based on sources from national organisms and on literature in the national context.

Finally, in the form of research-based design, the study proposes a design guideline to integrate the UGI approach in the given historical morphologies by implementing the retained historical aspects of the site, from the one hand, and detecting the forms of GI elements that can be applied in the site, from the other hand.

## 3. Urban Green Infrastructure Approach, Applicability, and Benefits

The UGI represents a complex system of natural and semi-natural areas and urban spaces, providing a wide range of valuable ecosystem functions and services for human environments [8] (pp. 132–138). GI responded to several challenges in urban planning principles, from environmental, economic, and societal points of view. The impact on the environment is perceived through the green infrastructure capacity to improve the quality of air, water and soil, regulating the climate, mitigating noises, and promoting biodiversity [9] (pp. 227–241). Social benefits are related to the wellbeing of the inhabitants and the connection of city users with the urban green spaces, which can promote cultural activities and social interactions [5–10]. In certain areas, contextual challenges should be taken into consideration as well, such as safety concerns [5]. Green infrastructure also contributes to the strengthening of the connectivity with nature, heritage, and place [9] (pp. 227–241). From an economic point of view, this approach can bring more value to the site, attracting more people for visiting green areas and hence, living in a healthier environment; besides, GI approach may initiate various investments. In a dense urban fabric, at the mezzo and micro scales, small interventions such as tree planting, providing planters and greening roads can increase surface permeability, contribute in improving local amenities, reducing flooding, controlling the local temperatures, increasing economic value, and improving community health and wellbeing [9] (pp. 227–241). The elements of UGI can be multiple, and their application depends on the addressed contextual specific issues [11] (pp. 1127–1142).

The identified types of green infrastructure are various and multiple; however, there is a general consensus about the diverse nature of GI in terms of size, function, and space vocations [12] (pp. 12–17), [13]. The most common GI elements used in urban fabrics, in streets, and open spaces are urban trees and green alleys. In buildings and buildings' envelopes, the often used GI elements are green roofs, roof gardens, vertical green systems, planter boxes and hanging pots, bird boxes, and roost sites. In open space types, the common managed landscape elements are permeable vegetated surfaces, public parks, community gardens, and unmanaged green sites like vacant lots [9] (pp. 227–241), [11] (pp. 1127–1142), [14]. Other examples can be explored too, depending on the scale and typology of the areas of intervention [9] (pp. 227–241).

In the Maghreb region, national and local organisms need to define strategic guidelines to adapt the UGI method in regions with particular and unique urban landscape characters. In fact, despite the multiple benefits of the green infrastructure adaptation in urban areas, related to human wellbeing and urban structure improvement, the different representations of the green infrastructure (GI) term didn't lead to a uniform process of this concept implementation [10]. On the other hand, researches about green components of Medinas urban landscapes in the Maghreb region are usually concerned only with the historic gardens outside the city. However, only few studies have been conducted about the improvement and the implementation of green elements inside these cities.

## 4. Islamic Medinas' Urban Development and the Urban Landscape's Consequences

One of the most valuable contributions of Muslim civilizations is their ingenuity in designing towns that can thoroughly adapt to the immediate environment, and the richness

of the green vocabulary related and adapted to their mystical ideology and culture, climate, and site conditions [15] (pp. 26–39). In North Africa, Muslim tribes emigrated from one city to another to establish their empires, leaving a valuable heritage. This chapter will first present the general urban characteristics of the studied Medinas and explore the relationship between the green elements and the historical centers.

As for the standard features of the two Medinas, the study highlights the characteristic of the organic urban fabrics. The cities' urban morphology consists of an assemblage of houses' courtyards (patios) linked by a hierarchical street network. Residential houses have massive facades with few openings, yet the courtyards connect the houses with the outside environment [16] (pp. 1–16). Moreover, while the functions are organized according to religious, social, and economic hierarchies, the urban system is not defined by uniform directions [3]. The width of the streets is not identical; the perspective is not direct in the public spaces as we discover the urban landscape sequence by sequence. This unique urban organization not only preserves the intimacy of the people passing by, but it also strictly protected the inhabitants from strangers who will find themselves lost in the labyrinth town [3]. The cities structure was mainly dictated by the religious and socio-cultural value systems related to relevant concepts like intimacy, privacy, ethics, hierarchy, and respect, which formed a sort of social convention that ruled the urban planning. That resulted, with other intervening factors, in the sprawling of narrow alleys, the density and proximity of the buildings, the enclosure/opening ratio of the facades, and water and vegetation presence characterizing the Islamic model of cities [3]. The social structure is also a determinant fact in shaping the Islamic city model, usually big families lived in the same house and shared the same courtyard and common spaces, however nowadays, the social structure and habits have changed.

*4.1. The Medina of Tunis*

The Medina of Tunis was founded in the 7th Century by Arab tribes. It covers around 290 Ha and has more than 700 monuments [17]. Its Souks, urban fabric, residential quarters, and monuments are the most protected and conserved in the Islamic world cities [18] (pp. 129–141). The city of Tunis was ruled by and hosted different civilizations. Hence, each empire marked the city with specific architectural and urban features [19] (pp. 8–25). On the other hand, green elements have played an essential role since the first dynasties. Farms and cultivated lands, also cemeteries, were outside the city fortresses and gates.

For the city's urban core, very few spaces in the Tunis Medina and its two suburbs deserve the denomination of green spaces [20] (pp. 18–31). As the present-day urban structure seems to exclude extensive green areas, the question is whether there were any green elements in the historical city and if they had meaning for the population.

The painting (Figure 1) illustrates the landscape of the city of Tunis in the late medieval era, it is directly accessible from the lake of Tunis and surrounded by several elevated hills. Both inside the city core and outside the Medina walls, diverse solitary plantations of palm trees appear. The enclosed agricultural lands must have been fertile by various species. The picture shows several castles with private gardens too.

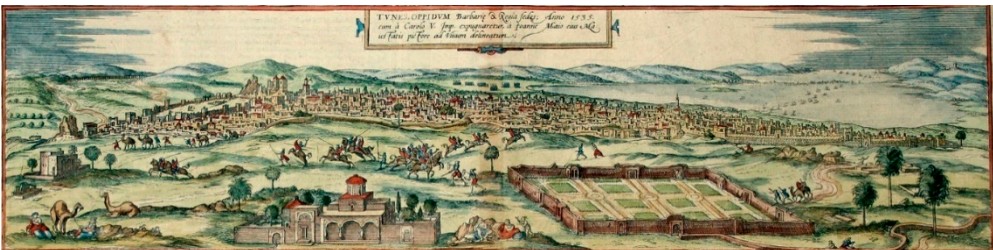

**Figure 1.** An overview representative image of the Medina in the 16th Century (data source: Atlas de Braum, 1575).

The below illustration of Figure 2 shows an abstract drawing of the Medina of Tunis's first core and the beginning of its urban area extension. It also illustrates the different gates and castles in the peripheries of the town.

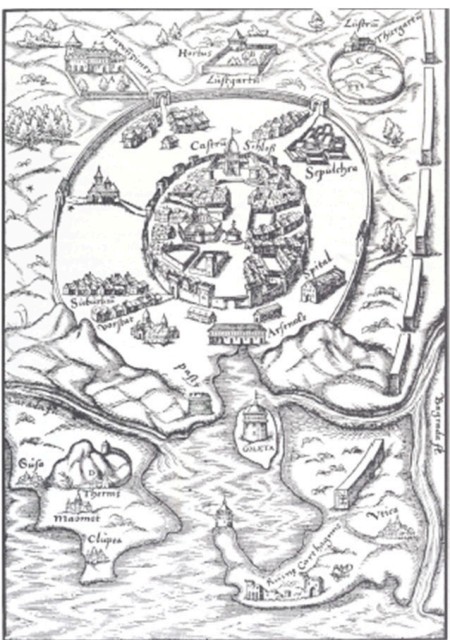

**Figure 2.** Tunis city from an engraving German of a book not identified from the 17th (data source: Abdelkafi, J. *La médina de Tunis, espace historique*; Presses du CNRS: Paris, 1989).

Castles with distinguished gardens of wealthy and governing members of the society appeared in the suburbs only after three centuries of city development. These estates have primarily started with the "Aghlabid" dynasty (800–909 AC). Yet, the most preserved castles are from the "Hafsid" period (1228–1574 AC), [19] (pp. 8–25), and the newer palaces belonged to the "Ottoman" leaders (Beys). The Ottomans masters' primary concern in the design of Islamic gardens was to give attention to every emanated detail; this had gone as far as the obsession of expressing the monumental, the beautiful, and the sumptuous [21].

Shrubs and trees were used for ornamenting houses' facades or streets (see Figure 3) in the green vocabulary inside the dense Medina core. Patios are usually abundant with plantations and might have a central fountain or tree. Nowadays, along with the evolution of the historic city, the tree is becoming an element more and more present in the public spaces [16] (pp. 1–16).

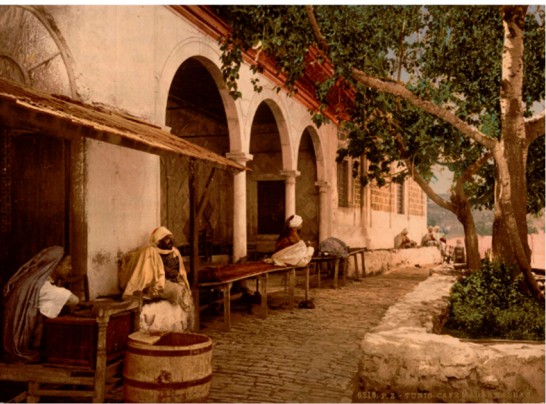

**Figure 3.** Traditional coffee place, Tunis, 1899 (data source: Bibliotheque numérique mondiale: https://www.wdl.org/fr/ (accessed on 7 January 2020)).

On the other hand, the appellation of several streets, squares and gates with reference to botanical meanings can indicate the importance of the vegetal elements in the inhabitants' lives. The Medina of Tunis was built around the "Zitouna" Mosque, "Olive Tree" Mosque. Other instances can be mentioned too, such as, "Bab El-Khadhra" gate meaning the "Gateway of green" (it was connected to agricultural lands from the Carthage site to the Medina). Also, "Bab El-Alouj", which used to have the name "Bab El-Erahba", signifying a small esplanade gate because it was the link between the "Kasbah" fortress and the royal parks ("Ras-Tabia" and "Abou-Fihr"). Several streets got their names after trees, such as palms, grenadines, figs, and plums. Two explanations are most likely; the first is that the city extended its territory on orchards and other gardens planted with these species, and the other justification is that during periods of depopulation or ruin, certain species might have reappeared [16] (pp. 1–16).

The city of Tunis started to acquire a different urban fabric, especially after the French colonizers influenced the Bey leaders to impose their urban and architectural character. After 1860, the difference between the urban fabric of the Medieval city and the Modern one became recognizable [3] (as shown in Figures 4 and 5), and especially, the urbanization progression that started since many people moved to the capital for better life conditions, was in the detriment of the surrounding green areas around the Medina [22] (pp. 23–30).

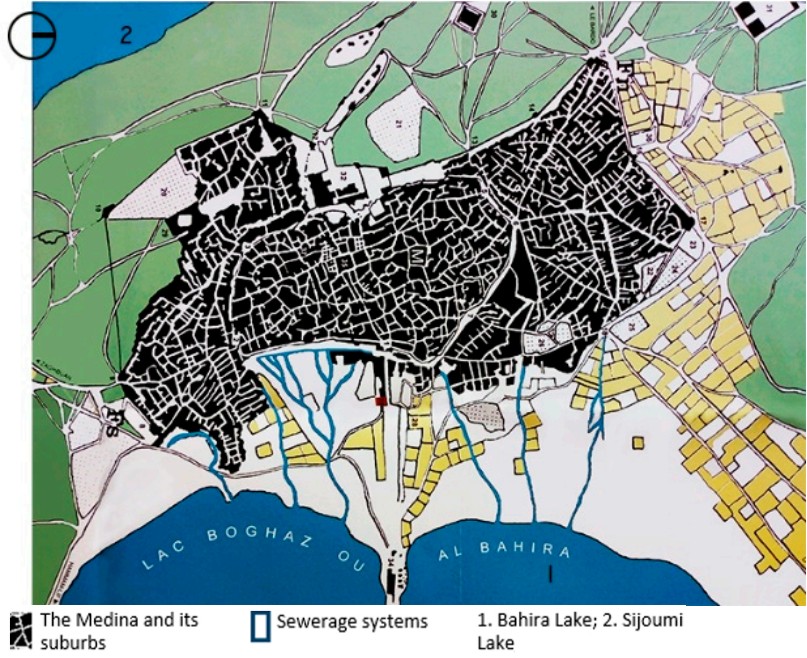

**Figure 4.** The Medina of Tunis and the beginning of the appearance of surrounding urban forms, in 1860 (adapted data source: Ammar, L. Les enjeux du patrimoine ancien et récent à Tunis aux XIXe et XXe siècles. Al-Sabîl: Revue d'Histoire, d'Archéologie et d'Architecture Maghrébines, 2017, 03).

During the French protectorate, Tunis has witnessed a European type urban development with a grid plan around the ramparts of the Medina, followed by the creation of several squares and green areas, among which is the Belvedere Park (1892, 100 Ha). The English landscape style park is not just connected with the colonial city but has a link also to Medina entrances ("Bab El-Khadhra" and "Bab El-Assal") [23] (pp. 7–23). Moreover, several tree lines were planted around the peripheries of the city. In this period, numerous public parks and urban alleys appeared in the city's surroundings to offer more recreation and ecological services; on the other hand, the French urban designers wanted to showcase and affirm the western identity and impress the indigenous inhabitants through their strategic realizations [23] (pp. 7–23).

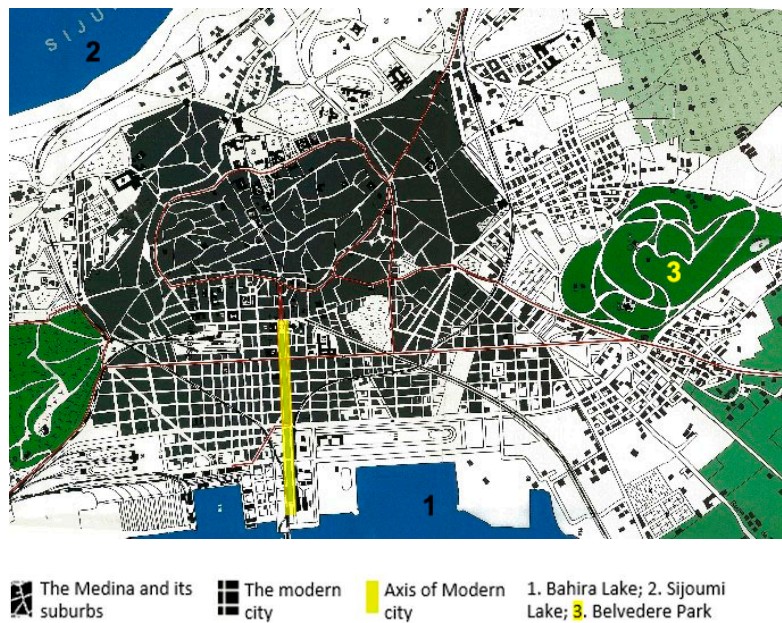

The Medina and its suburbs   The modern city   Axis of Modern city   1. Bahira Lake; 2. Sijoumi Lake; 3. Belvedere Park

**Figure 5.** The Medina of Tunis and the Modern city development along an urban axis that starts from the Medina, in 1930 (adapted data source: Ammar, L. Les enjeux du patrimoine ancien et récent à Tunis aux XIXe et XXe siècles. *Al-Sabîl: Revue d'Histoire, d'Archéologie et d'Architecture Maghrébines*, 2017, 03).

*4.2. The Medina of Marrakesh*

The Medina of Marrakesh, a UNESCO World Heritage site since 1985, covers around 700 Ha [24]. It was founded in the 11th Century by the "Almoravid" dynasty. The land that they chose was near the Atlas Mountains and rivers from which they sought water supplies. The site had a sacred significance thanks to its "Amazigh" divinity protection [25] (pp. 144–156).

The Marrakesh urban plan consists of a hierarchical diagram based on successive concentric circles and constructions surrounded by orchards and cultivated farms [26] (pp. 193–201). The distribution of green elements fitted into this organizational scheme. Green areas (large natural areas or isolated trees and plants in front of buildings facades) belonged to the compact urban core. Gardens and all green infrastructure elements, such as broad public and regional parks, surrounded the "great" city (the urban center in Figure 6), and farmlands used the last bordering fringes of the town [26] (pp. 193–201).

The "Almoravid" Muslim tribes settled in a site to establish a new city in the image of a "garden city" (The term "garden city" here does not refer to the Garden city movement initiated in the end of the 19th c by Ebenezer Howard in the United Kingdom. The authors refer to this expression as it was used in the book chapter (El Faïz, M. Marrakech: An Ecological Miracle and its Wanton Destruction. In: Gardens, City Life and Culture. Conan, M.: Whangheng, C.: *Gardens, City Life and Culture*: Harvard University Press: Scotland, 2008, pp. 193-201).) and showcase their ingenuity in landscape art and design. "The model of the garden city prevailed everywhere. From Damascus to Baghdad and from Cordoba to Marrakesh, the houses resembled rectangles and cubes engulfed in a sea of greenery" [26] (pp. 193–201). Thus, the art of gardens, which is among the most precious cultural goods of the Islamic civilization, is accompanied by the Maghreb essential cities [27] (pp. 87–90).

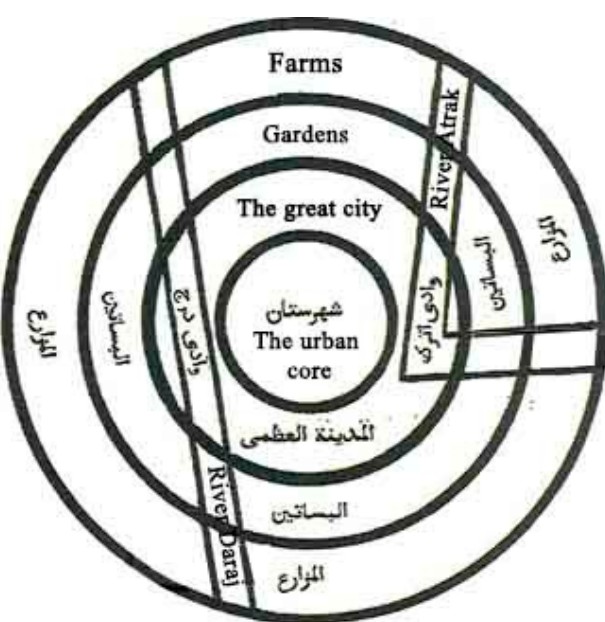

**Figure 6.** The urban structuring of the Medina of Marrakesh inspired by Qazvin diagram (data source: El Faïz, M. Marrakech: An Ecological Miracle and its Wanton Destruction. In: *Gardens, City Life and Culture*. Conan, M., Whangheng, C.: Gardens, City Life and Culture: Harvard University Press: Scotland, 2008, pp. 193–201).

As represented in the Figure 6, "the thirteenth-century geographer Al Qazwînî (active around 1275) drew a diagram of the garden city based on Qazvin, his place of birth, now located in Iran. Here one can see three concentric circles and a nucleus of buildings surrounded by orchards and cultivated plots" [26] (p 194). Also, castle gardens and farms are located within the city limits and constitute a homogeneous urban entity with the rest of the town (see Figures 7 and 8) [27] (pp. 87–90).

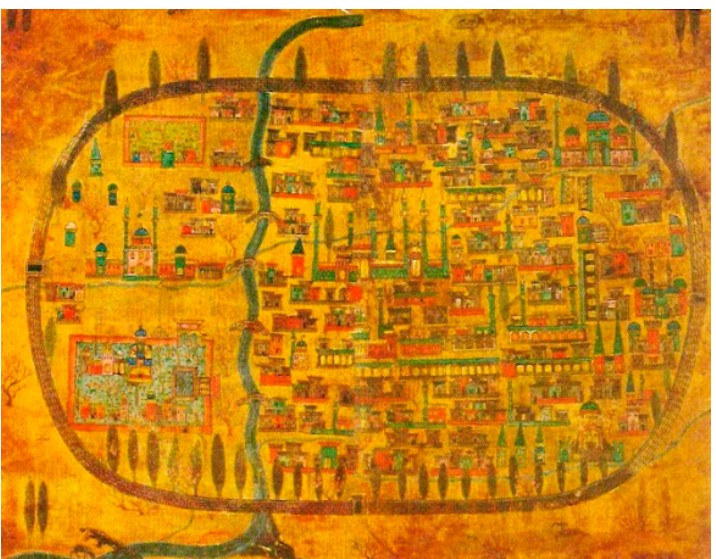

**Figure 7.** The green system structuring of the Medina of Marrakech. (data source: El Faïz, M. Marrakech: An Ecological Miracle and its Wanton Destruction. In: *Gardens, City Life and Culture*. Conan, M., Whangheng, C.: Gardens, City Life and Culture: Harvard University Press: Scotland, 2008, pp. 193–201).

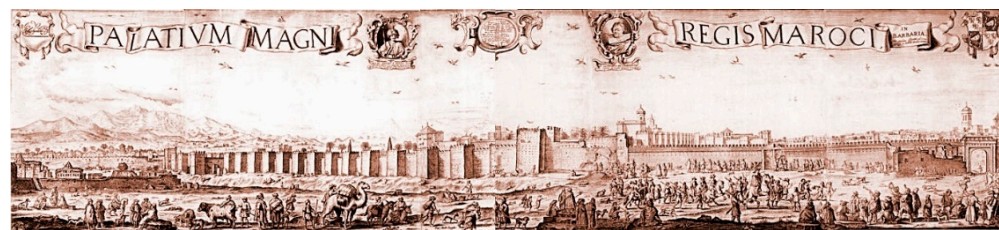

**Figure 8.** The Medina of Marrakesh in the Medieval period (data source: A drawing of the Badi palace in its heyday, by Adriaen Matham, 1640: https://landscapelover.wordpress.com (accessed on 20 December 2019)).

Extensive open lands followed the slopes and overlooked the Medina of Marrakesh. Therefore, the mastery of hydraulic and water techniques was essential in Islamic landscape design. Draining galleries (Khettaras) inherited from the early Islamic period techniques and aqueducts used to transport water [28] (pp. 11–13) with large water reservoirs and supported the irrigation too, for drinking water and sociocultural activities and festivities.

The open lands fit smoothly into the natural specificities of the site. These landmarks were carrying the Berbers culture by commemorating their pasturage. The most popular garden type was the "Agdal" (belonging to sultans and nobles of the city and associated with substantial mansions, palaces, or houses, located on the margins of the Medina or inside it). The "Agdal" garden model came from the "Almohades" in the 12th Century and appeared in Seville. The language of gardening unified the two realms from opposite sides of the Mediterranean [27] (pp. 87–90).

Diverse gardens with different styles are also inherited from this era, for instance, Arsa, Jnân, Riyâd. All these creations belong to the most valuable living heritage of the Muslim culture, evoking their utopia for an ideal world and the reverie of the heavens as described and idealized in their holy book.

Later on, during the French colonization that officially took over the majority of the country of Morocco, including Marrakesh, in 1912 (while Spain controlled the South-Eastern part of the country), the French architects brought significant transformations to the city's urban landscape. A European type of city following a regular-grid urban morphology broke the urban traditions (see Figure 9). The figure below shows the contrast between the Medieval and 20th Century plans [29] (pp. 337–349).

One of the new urban organization's purposes was to improve the urban climate and mitigate the urban heat by shading and evapotranspiration. Urban planners opted for an arcade system along the streets to favor refreshing air flows in the new districts. Also, tree alignments were fully integrated into urban planning. The urban atmosphere maintained a quality of life for the European populations. The colonial city, which constitutes a break from an urban point of view, had managed to integrate into its organization some of the old gardens with heritage value, such as "Jnan El Harti" and few others. The modern city included the implementation of plants in public spaces, which marked a difference from the garden city of the "Medina model", where the core city had mainly private greens, inside the houses and partly on the streets on the walls [30] (pp. 1–15).

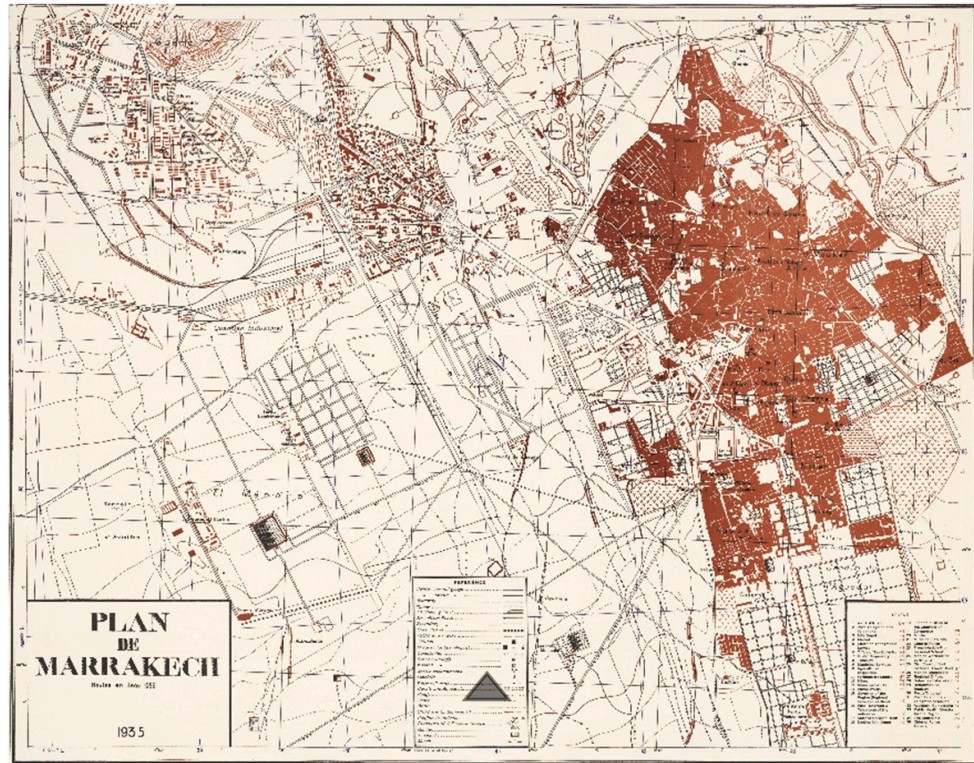

**Figure 9.** Map showing the contrasts between the Medina and the Colonial urban tissues, Marrakesh in 1924 (data source: https://thehouseinmarrakech.wordpress.com (accessed on 25 March 2020)).

*4.3. Comparative Analyses of the Two Cities' Urban Greenery*

We conclude from the cities' descriptions that the two Medinas have mutual characteristics in their general urban structure. Both cities exclude vast green spaces inside the urban core. However, in the case of Marrakesh, designed castle gardens and farms constituted a unified urban entity with the rest of the city's habitats and equipment. While in Tunis, we find such planned green areas a few miles away from the city.

These differences can be explained by the site conditions and topography. Furthermore, the arid climate conditions in Marrakesh called for cooling and temperate structures as green and blue areas and realizations. Yet, the adequate explanation behind the distinct green infrastructure of the two Medinas is the differences in their history; the Medina of Tunis was founded more than three centuries before Marrakesh even though that both cities were ruled by "Almohades". After conquering the Moroccan town from the "Almoravid" in 1147, this dynasty ruled the city of Tunis for around seventy years (1159–1227 AC). Their first architectural realization in the Medina of Tunis was the fortress "Kasbah". This monument was reflecting at that time the union between "Ifriqiya" and the extreme Maghreb. However, the "Kasbah" of Tunis differentiates from the one in Marrakesh by the absence of gardens [19] (pp. 8–25). Later on, and since the colonial period, the two towns had faced similar circumstances; both cities have experienced a rapid urbanization process to the detriment of the surrounding green lands.

The study suggested recalling the geomorphological and urban features that the two Medinas have in common and emphasizing the Islamic landscape and garden culture values. The Islamic landscape design and traditional methods were primarily considering the adaptation to the climate conditions, as the water supply, the use of sustainable irrigation systems, and open space perspectives in the landscape. Furthermore, Islamic green spaces such as "Agdal" gardens referred to the natural landscape of the indigenous civilizations and evoked the Berbers green pasturages. In addition to that, sensory and spiritual circuits are the main events in Islamic gardens. In most cases, each closure got its unique character and is defined by a tree type. The meaning and the sensory effects of the trees are contin-

ually changing [27] (pp. 87–90). These specific landscapes are typically quiet places for self-contemplation and awareness of the purpose of the creation [31] (pp. 11–19). "( . . . ) There appear to be also four influencing factors for the Islamic gardens of Paradise, which are the Holy Qur'an, hadiths, arid environment, and the earlier civilizations. Nevertheless, the elements are much detailed taking into account every aspect of Paradise described by the divine sources" [32] (p. 177).

## 5. Discussion

The Medina of Tunis seems to have ecological aspects that provide users with micro and mezzo climate comfort. Indeed, referring to an experimental study led by Tunisian researchers comparing the Medina's thermal comfort and Tunis' colonial and contemporary urban plans, the Medina's compact urban tissue proved its advantages in the thermal comfort aspect, and the other studied urban morphologies offered inferior thermal comforts [33] (pp. 689–700) (Research led by Tunisian researchers in the Ecole National d'Architecture et d'Urbanisme of Tunis in 2015 which proved, by quantitative analysis, and after a comparison between the Medinal, colonial, and contemporary urban tissues, that the urban tissue of the Medina offers the most convenient thermal comfort from physical, psychological, and physiological aspects.). Nevertheless, in the last two centuries, the urban extension in Tunis and Marrakesh was to the detriment of green and agricultural areas. Climate issues, population increase, rural exodus, centralization of administrative and public amenities, the concentration of economic and industrial activities, public policy in land-use planning and land tenure are all constituting, at varying scales, the engine of this complex urban extensive growth [34] (pp. 431–447), [35] (pp. 1–31).

### 5.1. Urban Green Infrastructure and Local Climate Problems

For the two case studies, despite the efforts made by stakeholders since the colonization until the present time to create public green areas around the historical studied centers, the atmospheric conditions inside the Medinas are a problem that still appears poorly controlled. The Medinas' inhabitants, nowadays, are using modern cooling equipment in the hot seasons. Additionally, the streets' climate is becoming warmer, humid, and even unpleasant [35] (pp. 1–31), [36].

In Marrakesh, still, a few green open spaces are present inside the urban fabric. However, no protection policy has fundamentally included the natural or green areas of high ecological supply. Therefore, they remain threatened by the excessive pressure exerted by anarchic and disproportionate urbanization. The construction of the "Sidi Ghanem" district, for instance, happened on a 36 Ha large palm tree land. Similarly, hundreds of hectares of irrigated lands with valuable ecosystem supply disappeared, giving space for public housing developments in the west part of Marrakesh [37]. Most of the gardens located inside the ramparts had disappeared, except for the royal gardens and some "Agdals". That was sadly the beginning of the Moroccan "garden city" model vanishing [26] (pp. 193–201).

In the last decades, to confront the rapid urban sprawl toward the peripheries of the capital and improve the climatological issues, Tunisian urbanists and stakeholders have adopted a strategy of implementing several urban green areas. The National Agency of Environmental Protection has launched the National program of urban parks, which is one of the most ecological actions that promote the development and implementation of urban parks in the city [38]. Unfortunately, many of these studies are still on papers. Moreover, in the Urban Development Plan of the municipality of Tunis, the Medina is listed as a unified zone with multiple architectural regulations mainly, but plans to increase the urban greenery are limited to the outer part of the Medina [39] (In the Urban Development Plan (Plan d'aménagement Urbain (PAU)) of Tunis, the green zones are marked in green only in the surrounding areas of the Medina, this latest is marked as a Medinal tissue zone, and in the PAU of the Medina of Tunis, there is no zoning for the green spaces in the

historical core.). Hence, the green network of Tunis is disconnected and unsatisfactory to the inhabitant's needs.

The two maps (Figures 10 and 11) represent the ratio of green spaces inside the UNESCO-protected Medinas zones (circled in red). Firstly, according to the maps' scales, we can observe that the surface of the Medina of Tunis represents only nearly 41% of the surface totality of Marrakesh Medina. Secondly, we notice that the ratio of green spaces compared to urbanized surfaces in the Medina of Marrakesh is higher than in Tunis Medina. In Marrakesh Medina, the ratio of green areas compared to urbanized zones is approximately 40%, while for the Medina of Tunis the ratio is around 12%; that can be explained with the absence of gardens in the Tunisian Medina protected zone, while in Marrakesh the Agdal gardens are covering the majority of green surfaces in the protected zone. In fact, in Tunis, The Medina and its environs (downtown area) are marking significantly higher temperature indications in the urban heat island maps of the city [40] (A study led by Tunisian researchers in cartography about the urban heat island in Tunis It has proved that the downtown of Tunis is marking significant difference comparing to some other areas of the Tunisian capital according to urban heat measurements).

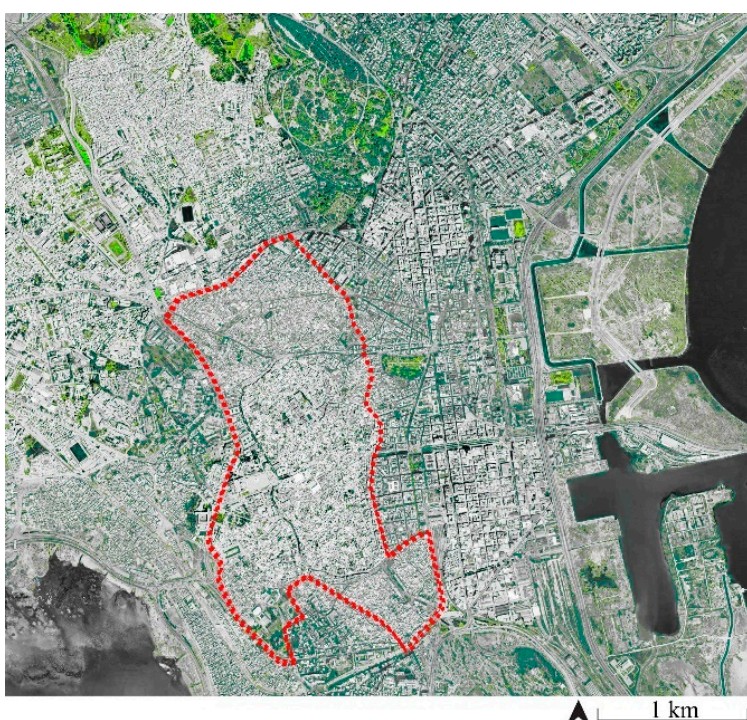

**Figure 10.** The UNESCO heritage site delimitation of Marrakesh Medina and the green spaces in the territory (data source: Authors).

*5.2. Urban Green Infrastructure Resolution in the Medinas*

Because of the loss of green spaces around the Medinas, and the extensive urbanization around the cities, the historic urban cores can be included under the theme of Sustainable Cities and Communities of the Sustainable Development Goals. Several nature-based and sustainable solutions can be considered in this regard. In fact, in the Medinas of Tunis and Marrakesh, there have recently been several initiatives to make the urban cores of the Medinas greener.

In the Medina of Tunis, The National Agency of Environmental Protection has included tree and landscape ordinances in plans and design guidelines down to the choice of materials and diverse ecological solutions [38]. Moreover, in the recent years, social initiatives appeared to empower the local community to implement active interventions and raise awareness towards the heritage of the Medina of Tunis. These actions aim to conceive

green elements in different areas of the Medina and create inclusive public spaces where the community will participate in setting up the project (see Figures 12 and 13) [41,42].

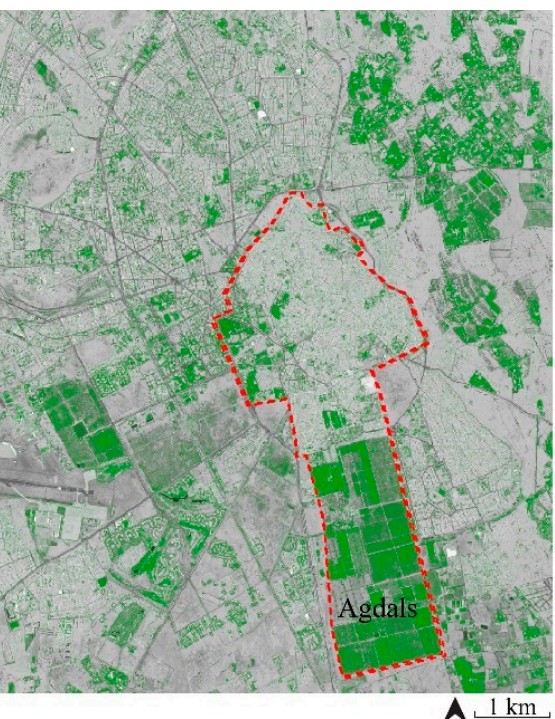

**Figure 11.** The UNESCO heritage site delimitation of Tunis Medina and the green spaces in the territory (data source: Authors).

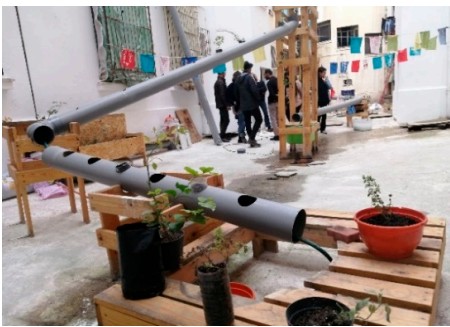

**Figure 12.** The project idea "Jnina Fel Medina" in a House rooftop in Hafsia quarter in the Medina of Tunis (data source: Faika Bejaoui (one of the project initiators) photography).

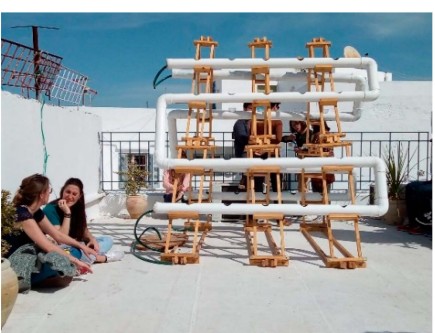

**Figure 13.** The project idea "Jnina Fel Medina" in a House rooftop in Hafsia quarter in the Medina of Tunis (data source: Faika Bejaoui (one of the project initiators photography).

For the case of Marrakesh, environmental actions aimed to ensure increasing sustainability for the future. Green spaces inside the Medina impacted by urban pressure needed protection and consideration. The Global Diversity Foundation has launched the initiative "For More Green Spaces in the Medina" to ensure the conservation of Medina's remaining trees. Several environmental and conservation projects carried out in Marrakesh increased urban ecological quality. The Urban Garden Project of the Ibn Abi Sofra School, located in an old, abandoned Agdal Bahmed Garden, involved different associations that collaborated to restore orchards and gardens over a one-hectare area [35] (pp. 1–31).

These initiatives should undoubtedly be reinforced and encouraged. On the other hand, stakeholders, local and governmental organisms are important actors in green infrastructure planning because that could involve new planning processes, knowledge, and other resources [10]. However, these initiatives are not well involved under official conceptualized regional schemes, and need to be strengthened by a theoretical framework.

*5.3. Research-Based Design for Urban Livability Improvement*

There are several key concepts in the application of UGI; however, the idea is relatively new [43]. In the case of historic dense fabrics, the application of this approach should be tested.

To illustrate the ideas that have been concluded from this study, an area from the northern part of the historical core of the Medina of Tunis was selected to define potential sites for green implementations. The study suggests to implement design based concepts in the sample area. The chosen section is in direct continuity with one of the most frequently visited circuits. Moreover, it is near the peripheral quarters of the Medina (El Hafsia), which represents a transition between the European and Islamic parts of the Tunis city. In the study area, several landscape elements which have the potential to host green elements were detected too (public squares, streets with various scales, and a private neglected educational institution).

The design proposition suggests to integrate green networks to improve the structural and functional connections among the small squares inside the city. In fact, linear elements such as trees and local species of shrubs can add more dynamism along the streets and reflect culture attachment feelings. These elements can improve the junctions between the public spaces but mainly help to strengthen the mezzo climate in the historical city. Another proposed solution is to add climbing and vertical green elements in the small public squares. Moreover, bushes and trees planted in movable planters and urban furniture can generate a greener and healthier environment, and vivid public life. (Kheireddine Museum, Hafsia Mosque and Romdhan Bey Squares in Figure 14). Terraces and rooftops can host diverse plant species, and private open spaces too (such as Israelite School Garden in Figure 14).

The Table 1 explains the design solutions in Figure 14. It indicates the detected space categories in the chosen study area, explains the potential green infrastructure elements that can be implemented, and proposes where these features could be applied. In the studied cases, the demonstration of the UGI approach can bring benefits to society and the environment, and it introduces different vital concepts. For instance, urban green elements suggest optimal solutions for space management and treating vacant spaces, too, especially in the context of historical sites, where some public spaces' functions and usage have been changed through time and are leftover. Additionally, these elements can enhance the quality of the open spaces and ensure green networking.

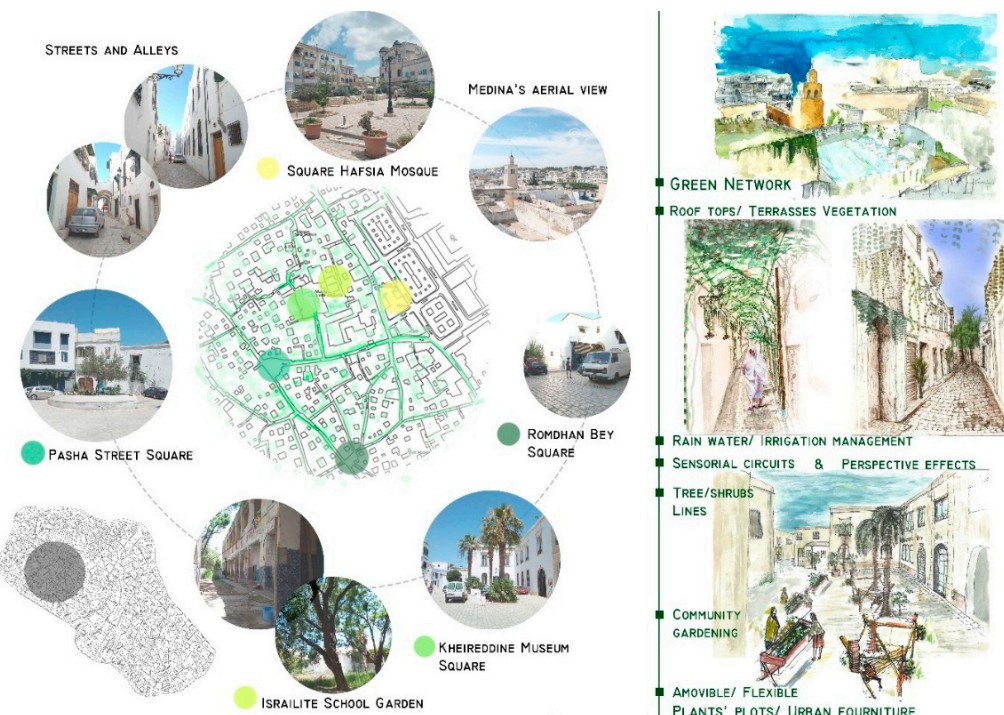

**Figure 14.** Plan and proposal to improve the selected area in the Medina of Tunis by implementing and enhancing UGI elements (data source: Authors photographs and sketches).

**Table 1.** Design proposals in the study area.

| Identified Landscape Elements in the Studied Area | Potential GI Elements | Aspects to Take into Consideration | Benefits to the Environment and Society |
|---|---|---|---|
| Public squares | -Movable urban tree pots. -Community gardens. -Landscape design in unmanaged green sites like vacant lots. | -Respect the urban morphology of the site -Respect the traditional buildings and infrastructure. -Take into account the fragility of the built heritage. -Apply planting design methods for semi-arid climate and the usage of native and drought-tolerant plant species. | -Promoting cultural activities and social interactions. -Improve community health and wellbeing. -Cooling local temperatures. -Strengthening the connectivity with the nature, heritage, and place. |
| Streets/alleys | -Shrubs planter boxes and hanging pots. | | |
| Buildings' rooftops | -Green rooftops. | | |
| Private educational institutions | -Permeable vegetated surfaces. | | |

## 6. Conclusions

The research highlighted that the Medinas' traditional urban fabric adapted well to climate conditions and social aspects of urban life for a long time in their history, resulting in the non-consideration of public green spaces' necessity in the city cores. On the other hand, the surrounding farmlands and fruit gardens were progressively exploited by the strict rebuilt settlement. Since the colonial period, several public parks and urban allées were implemented in the city's surroundings to offer more ecological and recreational services, and improve the urban character and landscape. However, public spaces in the Medinas and various streets and allées do not belong to these regional schemes as they are part of a firm historical context.

The comparative analysis of the two Medinas highlighted the common roots in the urban context, which means, at the same time, a profound coherence in aspects and means of problem-solving. The suggested UGI elements in the design guideline proposition could be adapted to the area's historic character and bring various benefits for society and the environment. Therefore, the landscape architecture tools suggested for the study area in Tunis Medina are possible and beneficial for the rest of Medina. Despite the different climate zone, the UGI tools may help effectively in climate change mitigation and adaptation.

The future research prospects are to consider surveys and data on urban heat island effects in the Medinas of Tunis and Marrakesh on the mezzo and macro scale of the two regions. On the other hand, to introduce the sustainability concept and techniques via urban development which undoubtedly follows the criteria of Medinas' livability. Therefore, the scope of this study is to raise awareness about climate change impacts on the urban heritage and to broaden the research and involve other Medinas in the Maghreb region and other historic cities with a dense urban fabric primarily.

In conclusion, the study intended to contribute to the understanding of distinguishing urban green infrastructure approaches in the Islamic historical urban areas by retaining the principles of these approaches and the Islamic landscape and garden culture values.

**Author Contributions:** Conceptualization: S.B.S.; Formal analysis: C.L.; Methodology: M.S. and K.S. All authors have read and agreed to the published version of the manuscript.

**Funding:** This research received no external funding.

**Institutional Review Board Statement:** Not Applicable.

**Informed Consent Statement:** Informed consent was obtained from all subjects involved in the study.

**Data Availability Statement:** Most of the used data are compiled from various publicly available sources or from author analysis.

**Conflicts of Interest:** The authors declare no conflict of interest.

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
