# Peer review of "Green System Development in the Medinas of Tunis and Marrakesh—Green Heritage and Urban Livability"

_2673-4834, doi:10.3390/earth2040048_

Round 1

Reviewer 1 Report

This study has tried to investigate and review the green infrastructures for the selected case studies. Although the manuscript is interesting, there are major corrections needed to be done to further improving the quality of the manuscript. It seems this manuscript is extracted from different parts of a Ph.D. thesis. This gives a feeling some parts of the text are missing. Thus, it is essential to do a major revision as suggested in follow:

  • Abstract

The abstract should be in a single paragraph. Add a sentence or two about the findings and importance of this study. To improve your abstract quality check some of the MDPI published articles.

  • Introduction
    • This section should be strengthened by adding recent studies related to green systems, vertical green systems, etc on an urban scale.  It should be more generalized so that has a wider range of audience rather than locals.
    • After the literature review, the aim, scope, and contribution of this study to knowledge should be explained in the last paragraph of the introduction, as well as the structure of this paper. Currently, the second paragraph explains the aims of this study as well as the last paragraph.
    • Line44, “Urban Green Infrastructure” change it to “urban green infrastructure (UGI)”. Follow this advice for all the abbreviations throughout the manuscript. Such as line 62, “GI”, etc. The full term should be given first time appearing in the text as well as short terms in a bracket
  • Methods
    • Overall, the methodology is so weak and needs to be rewritten entirely. “We analyze the urban development and transformation of the urban heritages” What kind of analysis has been done?  How was data collected? What are the justifications for chosen methods? Any photo, graph, etc to do some illustration for the study site? What is the basis for the design principal suggestions?
  • Section 3:
    • In general, this section has reviewed the history but, I found it difficult to understand the importance of this history review or the scientific aspect. What are the lessons that can be learned? What is the significance of this review? How the review can be useful to other researchers?
    • Figures 4 and 5 are not clear. Use each figure caption directly below the figure to check this through the manuscript.
    • What are the impacts of GI in the urban area of the cities?
    • Section 3.3: Is there any justification for this conclusion? “Two Medinas have mutual characteristics in their general urban structure” is this only based on the authors' deduction and the review that is made? Can you provide any supportive data?
    •  
  • Discussion
    • The findings of this study should be discussed and compared with other studies and cities.
    • What are the suggestions for the future of GI in these cities? How it can be improved?
    • Section 4.2, there is no need to give information on how this subsection is going to be procced in the first paragraph. This is redundant information.
    • Figures 12, 13, and 14 have not been cited in the text.
    • Section, 4.3 most of the suggestions here are based on the authors' opinion, there should be some justifications for each suggestion. Can use relevant studies citation to improve it. Particularly in the 3rd paragraph.
    • The explanation and justifications for the selected site should be given in the method section. “The selection of these open spaces is based on our perception and appreciation too.” This is not good enough for a scientific manuscript.
  • Conclusion
    • The conclusion should be rewritten. It does not address the objectives of this study and brings new ideas into the discussion.
    • What is the contribution of this study? How this study can be used for similar regions in different cities with various backgrounds?
    • What are the future studies?
  • Others
    • Avoid the first-person language in the manuscript
    • Follow up the Earth, MDPI format for the in the text citations.

Author Response

Response to reviewer 1 comments

The responses explain the modifications added to the article following the reviewer's suggestions. The answers and explanations are in red.

Remark: the numbers of chapters mentioned by the reviewer were changed because we included chapter 3. When the reviewer refers to chapter 3, now it is chapter 4.

Abstract

The abstract should be in a single paragraph. Add a sentence or two about the findings and importance of this study. To improve your abstract quality, check some of the MDPI published articles.

The authors added sentences (lines 14-20) to the abstract to better explain the objective of this study and its importance in the scientific domain.

Introduction

This section should be strengthened by adding recent studies related to green systems, vertical green systems, etc., on an urban scale.  It should be more generalized so that it has a wider range of audiences rather than locals.

The authors added chapter 3, which introduced the urban green infrastructure approach since it is part of the main concepts of the research. More arguments and scientific sources about the benefits of the urban green infrastructure on the environment, the social, and economic sectors were indicated too. After that, the authors provided more information and references about the usage of green infrastructure in an urban context by citing examples of green infrastructure elements according to different urban typologies and categories.

After the literature review, the aim, scope, and contribution of this study to knowledge should be explained in the last paragraph of the introduction, as well as the structure of this paper. Currently, the second paragraph explains the aims of this study as well as the last paragraph.

The authors added a paragraph about the scope and aims of the research at the end of the Introduction part. The general objective of the study is to bring into fore the idea of greening the Medinas. This problem emerged mainly because of urban changes during and after the French protectorate. The research aims to improve the quality of public spaces in the Medinas in the present context and demonstrate the application of UGI in a historic site with a dense urban fabric.

Line 44, "Urban Green Infrastructure" change it to "urban green infrastructure (UGI)". Follow this advice for all the abbreviations throughout the manuscript. Such as line 62, "GI", etc. The full term should be given first time appearing in the text as well as short terms in a bracket

The authors verified the abbreviations and wrote them according to the MDPI style guide.

Methods

Overall, the methodology is so weak and needs to be rewritten entirely. "We analyze the urban development and transformation of the urban heritages" What kind of analysis has been done?  How was data collected? What are the justifications for chosen methods? Any photo, graph, etc., to do some illustration for the study site? What is the basis for the design principal suggestions?

The authors modified the methodology part and mentioned the research materials.

Section 3:

In general, this section has reviewed the history but, I found it difficult to understand the importance of this history review or the scientific aspect. What are the lessons that can be learned? What is the significance of this review? How the review can be useful to other researchers?

(Now section 4): The importance of the history of case studies, their urban and green structure development was explained in the methodology and the research results parts. The conclusions from this study are important because they discover the values and attributes of the urban structure in the cities and the Islamic green heritage. It also proves that the green elements surrounding the Medinas have decreased.

Figures 4 and 5 are not clear. Use each figure caption directly below the figure to check this through the manuscript.

Verified.

What are the impacts of GI in the urban area of the cities?

The authors introduced the benefits of GI in urban cores in general in the 3rd chapter about UGI. A table was added too, Table 1, to explain these impacts.

Section 3.3: Is there any justification for this conclusion? "Two Medinas have mutual characteristics in their general urban structure" is this only based on the authors' deduction and the review that is made? Can you provide any supportive data?

(Now section 4.3): This is not new knowledge in the scientific domain. However, we have proved it too after the analysis and the comparison of the two cities.

Discussion

The findings of this study should be discussed and compared with other studies and cities.

The concept of green infrastructure inside the Medinas of Maghreb region is not a widely apprehended theme in the scientific domain, especially when it comes to a comparative analysis between different cities. In this study, the authors analyzed two Medinas. However, in future publications or researches, other Medinas examples can be studied too.

What are the suggestions for the future of GI in these cities? How it can be improved?

The authors mentioned in the discussion that the research prospects (in the future) are to consider surveys and data on urban heat island effects in the Medinas of Tunis and Marrakesh on the mezzo and macro scale of the two regions.

Section 4.2, there is no need to give information on how this subsection is going to be procced in the first paragraph. This is redundant information.

(Now section 5.2): We removed it from the text.

Figures 12, 13, and 14 have not been cited in the text.

They are cited in the text now.

Section 4.3 most of the suggestions here are based on the authors' opinion. There should be some justifications for each suggestion. Can use relevant studies citation to improve it. Particularly in the 3rd paragraph.

The explanation and justifications for the selected site should be given in the method section. "The selection of these open spaces is based on our perception and appreciation too." This is not good enough for a scientific manuscript.

(Now section 5.3): The choice of the studied area (within the Medina of Tunis) was justified and without a subjective argument this time.

Conclusion

The conclusion should be rewritten. It does not address the objectives of this study and brings new ideas into the discussion.

What is the contribution of this study? How this study can be used for similar regions in different cities with various backgrounds?

What are the future studies?

The authors mentioned the study objective and scopes in the discussion part, and stated in the conclusion that further studies should involve other Medinas in the Maghreb region and other historic cities, especially with a dense urban fabric. Also, the authors emphasized the intention of the study to distinguish the UGI application in Islamic urban areas by referring to the principles of this approach and the values of landscape and Islamic garden culture values.

Others

Avoid the first-person language in the manuscript

We verified that the first-person language is less used in this version of the paper.

Follow up the Earth, MDPI format for the in the text citations.

We have verified it.

Reviewer 2 Report

It can be publish 

Author Response

Response to reviewer 1 comments

We improved the research method and expounded the research material in more detail. The authors added a new chapter (chapter 3) to introduce the urban green infrastructure approach.

In the discussion part, we improved the results of the study. We also mentioned in the discussion that the research prospects (in the future) are to consider surveys and data on urban heat island effects in the Medinas of Tunis and Marrakesh on the mezzo and macro scale of the two regions.

Reviewer 3 Report

attached report

Author Response

Response to reviewer 1 comments

The responses explain the modifications added to the article following the reviewer's suggestions. The answers and explanations are in red.

Remark: the numbers of chapters mentioned by the reviewer were changed because we included chapter 3. When the reviewer refers to chapter 3, now it is chapter 4.

Having completed the review of the paper "Green system development in the Medinas of Tunis and Marrakesh - Green heritage and urban livability", I identified an interesting topic and valuable research, however, I suggest below some aspects:

Introduction:

Consider contextualizing the reader by briefly explaining ¿what is a medina?, since its definition may vary in different regions of the world.

We added a description of the Medina expression in the introduction part.

In the introduction, the authors mention important concepts such as ecosystem functions and services. "Will the cultural, historical, and landscape regulation functions be approached from the concept of ecosystem services?"

I suggest considering the theme of Nature-based solutions that will allow a better analysis, as well as the influence of climate change on the background.

In the corrected version of the paper, the study theme suggested (nature-based and sustainable solutions as responses to the urbanization and climate change issues) was mentioned in the abstract and introduction. The research structure is based now on nature-based solutions for the study cases as a response to the fast urban development in these sites and climate change impact on their urban core. The proposed method for nature-based solutions is the urban green infrastructure approach.

Materials and methods:

In this item mention is made of the urban cores or nodes, I consider that this is an aspect of great relevance in the analysis of the case study; therefore, it should be mentioned in the introduction. Additionally, it is not clear how these urban cores will be evaluated and how they will be selected.

I think you should be much more rigorous in describing your methodology, including references that confirm why you have selected particular approaches and relevance for this type of study. I suggest Yin's work by Yin, such as: Yin, R.K., 2009. How to do better case studies. The SAGE handbook of applied social research methods, 2, pp.254-282.

In the item "3.3. Comparative analyses of the two cities....." it is important to strengthen the comparative analysis of the urban cores identified above, taking into account that the authors give special importance to the description of the urban cores.

We suggest a research-based case study method to demonstrate the application of UGI in a dense urban fabric and in Medinas context in particular. We explain the study method based on case studies in the introduction part, particularly in the objectives and scope of the research.

Results

The authors have valuable information that could be better exploited, such as the description of the morphology of the urban fabric where I suggest to analyze that the transformation of this is given by the characteristics of the formation of the patios which strengthen human relationships and it is really these (human relationships) that transform the urban fabric.

We mentioned socio-cultural aspects that were the basis of the formation of the Islamic cities in the second paragraph of part 4.

The description in figure 4 and 5 is confusing.

Each figure is described separately now

In the item "3.3. Comparative analyses of the two cities....." it is important to strengthen the comparative analysis of the urban cores identified above, taking into account that the authors give special importance to the description of the urban cores.

It is possible that some information raised in item 4 as an outcome should be in this item.

Now Item 4.3: We added a paragraph (that was earlier in conclusion) to mention the retained values and attributes from the study of the Medinas' urban and Islamic green heritage characters. Also, we referred to these conclusions in the design proposition to mention that these values should be considered when applying a UGI approach and GI elements in the cities.

Discussion

Item 4.1 needs to expand and strengthen the information on local public policies.

(you refer to which is now Item 5.1): We mentioned the role of local and national policies and positions to implement green elements in the cities in the 5.2 chapter.

Item 4.2. could strengthen the discussion with an analysis from the conceptualization of ecosystem services and perhaps include sustainable development goals such as goal 11.

Now Item 5.2: We added a paragraph at the beginning of this chapter. We mention that because of the loss of green spaces and the extensive urbanization around the cities (as discussed and proved in the previous chapter), the studied historic urban cores can be included under the Sustainable Development Goals' Sustainable towns and communities.

Item 4.3. could optimizing the connection between urban centres improve urban liveability?

Now Item 5.3: Certainly, the connection between the different urban centres in terms of green infrastructure and urban integration will improve the urban liveability from various aspects, such as mitigating the problems of urban congestion and traffic around the Medina and improve the urban heat island impacts. However, even though we studied the history of the urban development of the cities and their connection with the modern French towns (to better frame the problems of the loss of the green elements around the Medinas), this study results concentrate on the applicability of green elements inside the urban core. Yet, the scope of this study for future analysis is to reflect the green infrastructure applicability from a larger scale.

Finally, in the introduction, the authors mentioned the concept of "urban pollution". However, this was treated ambiguously throughout the document; for the authors, what do they consider to be "urban pollution", it is important to leave clarity in the document.

The term of urban pollution was replaced by: urban heat island effects (rising temperature and degradation of mezzo climate). We removed the term urban pollution because the study focused more on the impacts of urbanization and climate change on the Medinas throughout history.

Reviewer 4 Report

Review, Manuscript ID: earth-1397384

“Green system development in the Medinas of Tunis and Marrakesh – Green heritage and urban livability”

Thank you for the opportunity to review this manuscript. It is a useful contribution to the literature on the historical development of green infrastructure in the context of Muslim civilizations and more importantly giving reflections on how the recent urban strategies related to the integration of green zones in the historic sites, and their feasibility can be used to inform policy decisions.

The paper examines and evaluates the development of green infrastructure in historical times as well as the contemporary urban landscapes in Medinas of Tunis and Marrakesh and highlights their tangible and intangible values. The paper is based on the analysis of urban development and transformation of the urban heritage, including the elements of green infrastructure, using historical literature as well as the historical and present maps. The paper also proposes a design guideline for the integration of the green infrastructure approach in the given historical morphologies by implementing the retained historical aspects of the case study areas.

The technical, structural, and formal quality of the manuscript is good. The manuscript is well written in terms of its coherence and well referenced.  Also, different sections are well balanced in length and contents.

The Materials and Method are thoroughly presented. The techniques for gaining information are clear and have been meticulously carried out. The results are thoroughly presented.

I enjoyed reading this paper.

I would only recommend that the final sections of this paper engage more with a broader literature on the planning and management of green infrastructure as a means to outline the contribution of the paper to our understanding of the influences on, and consequences of, differing urban green infrastructure approaches in the Muslim civilizations to the planning and management of different green spaces in urban areas as part of the Islamic landscape and garden culture values. 

Author Response

Response to reviewer 1 comments

The responses explain the modifications added to the article following the reviewer's suggestions. The answers and explanations are in red.

Remark: the numbers of chapters mentioned by the reviewer were changed because we included chapter 3. When the reviewer refers to chapter 3, now it is chapter 4.

I would only recommend that the final sections of this paper engage more with a broader literature on the planning and management of green infrastructure as a means to outline the contribution of the paper to our understanding of the influences on, and consequences of, differing urban green infrastructure approaches in the Muslim civilizations to the planning and management of different green spaces in urban areas as part of the Islamic landscape and garden culture values. 

We added other sections to the paper; Chapter 3 is about the UGI approach. It refers to the benefits and principles of applying this approach in the urban context, focusing on the urban areas with a dense urban fabric.

We mentioned the values of Islamic landscape and garden culture in the last paragraphs of the 4.3 section. We emphasised in the final sections (in the design guideline research proposal and the discussion and conclusion) the contribution of the paper to better understanding the facts which should be taken into consideration when implementing green elements in the context of Medinas Muslim cultures. We also mentioned the differing UGI implementation principles in the Medinas and similar cities from other urban domains.

Round 2

Reviewer 1 Report

I would like to thank authors for the major improvements, the manuscript readability and quality have improved significantly. The comments have been addressed well and I believe the manuscript will be ready for publication with only a few minor corrections as stated below:

  • Line 88, UNESCO, give the full term the first time it appears on the text as united nations educational, scientific and cultural organization (UNESCO).
    • Is not clear which Figures belong to which caption at Figures 10, and 11;  Figures 12, and 13. Use each figure caption directly below the figure.
    • There is two discussion section at this point, sec 5 and 6. Sec 6 should be with a different heading and perhaps a subsection to Sec 5.
    • The conclusion can be improved. Add a paragraph or two after the 1st one to indicate and explain the key findings of this study.

Author Response

Responses to Reviewer, second round revision

We are very grateful for the revision of the reviewer. We have made the changes that the reviewer asked for. The responses to the reviewers' comments in this document are in red colour.

The modification that we added in the paper are in Track change mode, please use All markup to see the changes.

o Line 88, UNESCO, give the full term the first time it appears on the text as united nations educational, scientific and cultural organization (UNESCO).

The authors have mentioned the full name of UNESCO as suggested in line 79 in the paper, because that is where the term has appeared the first time in the text after the abstract.

o Is not clear which Figures belong to which caption at Figures 10, and 11; Figures 12, and 13. Use each figure caption directly below the figure.

Verified. We made figures 6 and 7 separate too because the captions descriptions are similar.

o There is two discussion section at this point, sec 5 and 6. Sec 6 should be with a different heading and perhaps a sub section to Sec 5.

o The conclusion can be improved. Add a paragraph or two after the 1st one to indicate and explain the key findings of this study.

We modified the conclusion and removed the previous separate chapter of Discussion, the chapter 5. (Previously named 5. Discussion of the urbanization issues in the Medinas and study results) is now 5. Discussion because it involves the discussion of the research.

Other changes

  • We added a legend under Figure number 4, and modified the legend under Figure number 5.
  • We corrected the source number 26. It is a book chapter.
  • We verified the source is each figure's caption.
  • The citation written on lines 295 to 298 was verified and copied exactly as it is written in the original source, and the page number is added.
  • Minor grammar and orthograph verifications were done.

This manuscript is a resubmission of an earlier submission. The following is a list of the peer review reports and author responses from that submission.